# Molecular Hydrogen Modulates T Cell Differentiation and Enhances Neuro-Regeneration in a Vascular Dementia Mouse Model

**DOI:** 10.3390/antiox14010111

**Published:** 2025-01-20

**Authors:** Dain Lee, Hyunjun Jo, Jong-Il Choi

**Affiliations:** 1KU-KIST Graduate School of Converging Science and Technology, Korea University, 145 Anam-ro, Seoul 02841, Republic of Korea; sunkro@korea.ac.kr; 2Department of Neurosurgery, Korea University Ansan Hospital, Korea University College of Medicine, 123 Jeokgeum-ro, Ansan 15355, Republic of Korea; 3Department of Neurosurgery, Korea University Guro Hospital, Korea University College of Medicine, 148 Gurodong-ro, Seoul 08308, Republic of Korea; neurovascular.jo@gmail.com

**Keywords:** vascular dementia, molecular hydrogen, bilateral common carotid artery sclerosis, ischemia, antioxidant, reactive oxygen species

## Abstract

This study explores whether molecular hydrogen (H_2_) administration can alleviate cognitive and immunological disturbances in a mouse model of vascular dementia (VaD). Adult male C57BL/6 mice underwent bilateral common carotid artery stenosis to induce VaD and were subsequently assigned to three groups: VaD, VaD with hydrogen-rich water treatment (VaD + H_2_), and Sham controls. Behavioral assessments using open field and novel object recognition tests revealed that VaD mice exhibited anxiety-deficient behavior and memory impairment, both of which were reversed by H_2_ treatment. Histological examinations showed pyknotic neuronal morphologies and elevated reactive oxygen species (ROS) in the VaD hippocampus, whereas H_2_ administration mitigated these alterations. Furthermore, VaD-induced downregulation of BCL2 was reversed in the VaD + H_2_ group, in parallel with increased IL-4 expression. Flow cytometric analyses revealed that VaD disrupted T regulatory cell homeostasis by significantly increasing their proportion, an effect reversed by H_2_ treatment, thereby restoring immunological balance. Transcriptomic evaluations confirmed that VaD suppressed key neuroprotective and anti-inflammatory genes, while H_2_ treatment restored or enhanced their expression. Collectively, these findings highlight the neuroprotective and immuno-modulatory potential of molecular hydrogen, suggesting that H_2_ supplementation may promote neuronal resilience, modulate T cell differentiation, and support cognitive recovery in vascular dementia.

## 1. Introduction

Vascular dementia (VaD), traditionally recognized as the second most common type of dementia after Alzheimer’s disease (AD), has recently been suggested to be the most prevalent form of dementia among the elderly [1,2,3]. As the global population ages rapidly, the number of VaD cases is expected to rise, placing additional pressure on healthcare systems. Studies estimate that VaD accounts for approximately 15–20% of all dementia cases, and this percentage may be higher in regions with elevated rates of hypertension, diabetes, and stroke [4]. Furthermore, nearly 50% of stroke survivors may develop VaD-related symptoms within 25 years [5]. Additionally, cerebrovascular disease (CVD), the underlying condition of VaD, is the second leading cause of death worldwide and significantly contributes to cognitive decline and disability across populations [6].

VaD significantly impairs cognitive functions, including memory, executive function, language, and attention, thereby severely diminishing patients’ quality of life. Furthermore, VaD is associated with high mortality rates. Compared to AD, VaD is linked to a shorter life expectancy, with an average of 3.9 years from dementia onset to death, versus 7.1 years for AD [7]. Additionally, the chronic nature of VaD exacerbates mortality rates, especially when compounded by other CVDs, such as stroke [8].

From a societal perspective, the rise in VaD cases imposes substantial economic burdens on society. Health statistics indicate that VaD patients incur the highest annual costs among all dementia patients, including those with CVDs, as observed in New York, USA [9]. Direct costs include expenses for long-term care facilities and nursing homes, while indirect costs arise from reduced productivity, as patients and their caregivers may leave the workforce. As VaD progresses, individuals often require long-term care, increasing their dependence and placing considerable stress on caregivers. These financial pressures impact individuals, families, and national healthcare systems, with dementia-related expenses expected to escalate in the coming decades [7,10,11].

Despite its significant impact, treatment options for VaD are limited. Managing vascular risk factors like hypertension, dyslipidemia, and diabetes can help slow the disease’s progression, but there is no established cure or long-term therapy that can fully stop or reverse cognitive decline [12,13]. Current medications, such as acetylcholinesterase inhibitors, provide only temporary symptom relief. Antiplatelet agents may help to prevent further cerebrovascular events, but no drug has been proven to significantly improve symptoms once the disease has advanced [14,15,16,17]. Therefore, treatments for VaD have been mainly focused on controlling stroke risk factors.

Recently, molecular hydrogen (H_2_) has emerged as a promising therapeutic agent [18,19]. Studies have shown that H_2_ has anti-oxidative, anti-apoptotic, and anti-inflammatory effects in various conditions, including cancer, lung damage, and neurodegenerative diseases, by neutralizing ROS, which cause damage to DNA, proteins, and cells [20,21,22]. Given the significant role of ROS in exacerbating neurological damage, H_2_ presents a potential strategy for mitigating oxidative stress in dementia-related conditions. Regarding oxidative stress in disease, a VaD differs in its etiology from AD, as it occurs due to neurovascular dysfunction that induces ischemia or reperfusion, which can greatly increase the ROS that harm tissue. In contrast, AD primarily occurs through the accumulation of amyloid beta plaques. Therefore, managing the excessive accumulation of ROS in the brains affected by VaD constitutes a more critical challenge compared to AD [1,2,3].

In detail, H_2_ can exert its therapeutic effects on VaD through multiple cellular interactions at the molecular level. Primarily, H_2_ acts as a selective antioxidant, neutralizing harmful reactive oxygen species such as hydrogen peroxide (H_2_O_2_) and peroxynitrite (ONOO^−^), thereby reducing oxidative stress and preventing neuronal damage [23]. It also modulates inflammatory responses by suppressing pro-inflammatory cytokines and inhibiting transcription factors like NF-κB, which helps mitigate neural inflammation associated with VaD [20,22]. H_2_ further enhances the cell’s antioxidant defenses by activating the Nrf2 pathway, leading to increased expression of antioxidant enzymes [24]

Although hydrogen shows great potential, its effects on mammals, especially regarding immune-related mechanisms in neurodegenerative diseases, are not fully understood. Since neurodegenerative diseases often involve inflammation, understanding hydrogen’s role in immune modulation is crucial. Regarding this, we investigated the anti-oxidative effects of H_2_ in a mouse model of cerebral ischemia and reperfusion injury, demonstrating its ability to remove free radicals in our previous study published in 2021. We observed changes in cytokine levels, such as reduced IL-6 and increased IL-10 and IL-2, suggesting that T cells play a role in this process [18,22].

Building on our previous findings, the present study aims to elucidate the therapeutic potential and underlying mechanisms of H_2_ in treating VaD, with a specific focus on inflammation-related pathways. Given that VaD is characterized by chronic cerebrovascular insufficiency leading to neuronal damage, cognitive decline, and persistent neuroinflammation, effective treatment strategies must address both oxidative stress and immune dysregulation [25,26]. This study evaluates the impact of H_2_ on key inflammation-related factors involved in neuronal regeneration and regulatory T cell (Tregs) pathways using reliable techniques such as flow cytometry and polymerase chain reaction (PCR) analyses. We specifically measured alterations in cytokine levels, Treg populations, and the expression of genes associated with neuroprotection and antioxidant responses. Additionally, we assessed H_2_’s ability to reduce ROS accumulation. By providing comprehensive evidence of H_2_’s multifaceted role in mitigating oxidative stress and modulating immune responses, this research underscores the potential of H_2_ as a novel therapeutic agent for slowing the progression of vascular dementia and improving cognitive and neuronal health in affected individuals.

## 2. Materials and Methods

### 2.1. Animals

This study was approved by the Institutional Animal Care and Use Committee (IACUC) at the Korea University Medical Center (approval no. KOREA-2021-0176). We conducted a study using a male 30 g 12-week-old C57BL/6 mouse (DBL, Eumseong, Republic of Korea). Animals were group-housed with no more than 4 animals per cage and acclimatized to standard laboratory conditions on a 12 h light/dark cycle. Food and water were provided ad libitum.

### 2.2. Experimental Design

We conducted a study designed to establish the effect of hydrogen-rich water (HRW) on a mouse model of VaD and to analyze the mechanism of action of H_2_. Surgery to make a VaD model was performed on 33 mice and half of them were randomly selected and treated with hydrogen. Thus, the group was divided into an untreated VaD group (VaD; n = 15) and a hydrogen-treated VaD group (VaD + H_2_; n = 18), and this was compared with the Sham control group (n = 15). HRW was generated by the HRW machine (H2B-H20, Hommage, Yongchang, China). Hydrogen concentration in HRW was maintained above 1.000 ppm by refreshing it every morning until the end of the experiment. HRW was provided only to the mice of the H_2_ group instead of regular water and they were allowed to drink it ad libitum. In contrast, the Sham and VaD group mice were provided with regular water.

### 2.3. Bilateral Common Carotid Artery Stenosis (BCCAS) Operation

Anesthesia was induced in mice through inhalation of 5% isoflurane in 1 L of oxygen using an animal anesthesia machine (L-PAS-01D, LMS Korea, Pyeongtaek, Republic of Korea). Full anesthesia was achieved within 3 min. During the surgical procedure, anesthesia was maintained with 3% isoflurane in 600 mL of oxygen. A vertical midline incision was made along the ventral neck of each mouse, and the soft tissues and muscles were carefully dissected to sequentially isolate both common carotid arteries (CCAs). Both CCAs were occluded for 10 min using bulldog clips. To mitigate the risk of mortality due to reperfusion injury, the clips were removed at 3 min intervals. Subsequently, a 0.16 mm coil (SWPA 0.16, Sawane SPRING Co., Ltd., Hamamatsu, Japan) was wrapped around each common carotid artery to induce stenosis. After suturing the incision, the mice were housed individually until consciousness was restored, after which they were returned to their original cages. All procedures were completed within 30 min. This model was designed to mimic the conditions of VaD, characterized by reperfusion-induced damage and continuous cerebral hypoperfusion, by employing an occlusion method aligned with Bilateral CCA stenosis (BCCAS) operational protocols.

### 2.4. Open Field Test (OFT)

For cognitive function tests, mice were translocated to a dimly lit test room 30 min before testing and the testing chamber was sanitized with 70% ethanol. In the open field test (OFT), we placed the mouse in the center of the chamber (acryl, 50 × 50 × 30 cm^3^) and allowed it to move freely for 5 min. Recording was started 2 min after leaving the mouse in the chamber and the movement pattern was analyzed using ANY-maze software Version 7.48 (Stoelting, Wood Dale, IL, USA).

### 2.5. Novel Object Recognition Test (NORT)

In the novel object recognition test (NORT), mice were individually placed in an empty acrylic chamber (33 × 20 × 30 cm^3^) for habituation over 3 consecutive days, with each habituation session lasting 5 min. On the test day, each mouse was placed in the chamber containing two identical objects positioned in opposite corners and allowed to explore for 5 min. After a 2 h interval, one of the two objects was replaced with a novel object, and the mouse was returned to the chamber for a second 5 min exploration period. Using this approach, we assessed the mice’s ability to recognize familiar objects and their propensity to explore novel objects compared to familiar ones. The mice’s movements were recorded and analyzed using ANY-maze software.

### 2.6. H&E

Mouse brain tissue was sectioned to a thickness of 6 μm. Brain tissue slices were stained with hematoxylin and eosin (H&E). The slices were mounted on cover glasses using a quick hardening mounting medium (03989, Sigma-Aldrich, St. Louis, MO, USA). The slides were digitized using a Pannoramic Digital Slide Scanner (3DHISTECH Ltd., Budapest, Hungary).

### 2.7. DHE Assay

Brain tissue slices (6 μm) were incubated with 10 μM dihydroethidium (D7008, Sigma-Aldrich, St. Louis, MO, USA) for 1 h in the dark at room temperature. The slices were washed three times with 1X PBS, with each wash lasting 5 min. Cover glasses were mounted onto slides containing the brain tissue slices using an antifade mounting medium containing DAPI (H-1200-10, Vector Laboratories, Burlingame, CA, USA). The tissue slices were observed under a fluorescence microscope (Axiovert 200M, Carl Zeiss Meditec, Dublin, CA, USA). The DHE intensity relative to DAPI intensity was calculated for each region of interest (ROI) using ImageJ software 1.53a (National Institutes of Health, Bethesda, MD, USA).

### 2.8. Malondialdehyde (MDA) Assay

Mouse hippocampal brain tissues were harvested and homogenized in 1 mL of cold assay buffer from the TBARS Assay Kit (STA-330, Cell Biolabs, Inc., San Diego, CA, USA). The homogenized samples were then centrifuged at 10,000× *g* for 10 min at 4 °C. The resulting supernatant was transferred to fresh 1.5 mL microcentrifuge tubes and butylated hydroxytoluene (BHT) was added from a 100X stock to achieve a final concentration of 1X in each sample. Malondialdehyde (MDA) standards were prepared by serially diluting MDA in double-distilled water to concentrations ranging from 0 μM to 120 μM. Next, 100 µL of either the MDA standards or the samples were placed into separate microcentrifuge tubes, followed by the addition of 100 µL of SDS lysis solution to each tube. After thorough mixing, the tubes were incubated at room temperature for 5 min. Subsequently, 250 µL of TBA reagent was added to each tube and mixed well. The tubes were then incubated at 95 °C for 60 min before being rapidly cooled on ice for 5 min. Following incubation, the samples were centrifuged at 3000 rpm for 15 min at room temperature, and the clear supernatants were carefully transferred to new microcentrifuge tubes. For spectrophotometric measurements, 200 µL of each standard and sample was aliquoted into the wells of a 96-well plate and the absorbance was read at 532 nm.

### 2.9. Immunofluorescence Microscopy

Immediately after sacrifice, mouse brains were fixed by immersion in 4% paraformaldehyde (PFA) for 48 h at 4 °C. The fixed brain specimens were then thoroughly washed multiple times with 1X PBS and cryoprotected by soaking in a 50% sucrose buffer for 48 h. Frozen brains were subsequently sectioned into 6 μm thick slices and blocked using a 5% bovine serum albumin (BSA) solution diluted in PBS. Each section was incubated with a primary antibody against BCL2 (ab59348, Abcam, Cambridge, UK) or IL-4 (SC-53984, Santa Cruz Biotechnology Inc., Paso Robles, CA, USA) diluted in the 2.5% BSA solution for 1 h at room temperature. Following primary antibody incubation, sections were treated with a secondary antibody conjugated to Alexa Fluor 568 goat anti-rabbit IgG (F0257, Sigma-Aldrich, St. Louis, MO, USA) for another 1 h at room temperature. After antibody labeling, a mounting medium containing DAPI was applied to each section, which was then covered with a cover glass. Immunolabeled proteins were visualized and analyzed using a fluorescence microscope.

### 2.10. Flow Cytometry

For flow cytometry analysis of the Treg population, a Foxp3 staining kit (560133, BD Biosciences, San Jose, CA, USA) was utilized. Mouse brain cells were isolated and resuspended in FACS buffer. Cells were stained for surface markers CD4 and CD25 using fluorescently labeled antibodies. Following surface staining, cells were fixed and permeabilized with the BD Transcription Factor Staining Buffer Set for intracellular Foxp3 detection. Intracellular staining was then performed using an anti-Foxp3 antibody. Stained cells were analyzed on a BD FACSCalibur flow cytometer. CD4^+^Foxp3^+^ Tregs were quantified within the CD4^+^ population.

### 2.11. Quantitative Real-Time Polymerase Chain Reaction (qRT-PCR)

Mouse hippocampal brain tissue cells were lysed and homogenized in 1 mL Trizol reagent (T9424, Sigma-Aldrich) in Eppendorf microcentrifuge tubes for 5 min. Subsequently, 0.2 mL chloroform (366927, Sigma-Aldrich) was added to the sample, thoroughly mixed, and incubated for 2 min at room temperature. The sample was centrifuged for 15 min at 12,000× *g* and 4 °C. The colorless upper aqueous phase was transferred to a new tube and 0.5 mL isopropanol (278475, Sigma-Aldrich) was added to the sample and incubated for 10 min at 4 °C. The sample was centrifuged for 10 min at 12,000× *g* at 4 °C. The supernatant was discarded and the pellet was resuspended with 1 mL 75% ethanol. The mixture was vortexed and centrifuged for 5 min at 7500× *g* at 4 °C. The supernatant was discarded. The RNA pellet was air-dried and resuspended in 50 μL RNase-free water (W2004, Biosesang, Seoul, Republic of Korea). The isolated RNA was reverse-transcribed to cDNA using the High-Capacity RNA-to-cDNA Kit (4387406, Thermo Fisher Scientific, Waltham, MA, USA). cDNA was mixed with Accutarget qPCR Screening Kit primers (SM-0005, SM-0087, SM-0130, Bioneer, Seoul, Republic of Korea) and SYBR™ Green Universal Master Mix (4309155, Thermo Fisher Scientific) and amplified in a two-step process with a 55 °C annealing temperature using the QuantStudio 5 Real-Time PCR System (A34322, Applied Biosystems, Foster City, CA, USA).

### 2.12. Statistical Analysis

All results are expressed as mean ± standard error of the mean (SEM). Outliers were excluded by conducting an outlier assessment per every analysis. To compare differences among the three groups, a one-way analysis of variance (ANOVA) was performed, followed by post hoc analysis. Statistical analyses were conducted using Prism 9 software (GraphPad Software, San Diego, CA, USA). A *p*-value of <0.05 was considered statistically significant.

## 3. Results

### 3.1. H_2_ Restores Cognitive Function, Reducing Anxiety and Improving Memory in VaD Cases 

After bilateral common carotid artery stenosis (BCCAS), including 10 min common carotid artery (CCA) occlusion and stenosis, was induced in 12-week-old male C57BL/6 mice (Figure 1A,B), the animals received H_2_ treatment for 12 weeks. In the first week after surgery, the open field test (OFT) was conducted to evaluate locomotor activity and anxiety levels, and the novel object recognition test (NORT) was conducted in the 11th week to assess learning and memory dysfunction in the mice that underwent BCCAS surgery (Figure 1C–F). In the OFT heat map analysis, VaD mice spent a significantly longer time in the center zone compared to the Sham or VaD + H_2_ groups (Figure 1E). Tracking analyses revealed no differences in travel distance (one-way ANOVA; *p* = 0.7986; n = 14 for all) or mean speed (one-way ANOVA; *p* = 0.5263; n = 14 for all). However, entries into the center zone were significantly higher in the VaD group (n = 13) than in the Sham group (n = 12, Tukey’s multiple comparison test; *p* = 0.0083). This tendency was reversed in the H_2_-treated group (n = 14) compared to the VaD group (*p* = 0.0475), with no difference between the Sham and H_2_-treated VaD groups (*p* = 0.7271). Similarly, latency to center entry was significantly lower in the VaD group (n = 11) than in the Sham group (n = 9, unpaired *t*-test with Welch’s correction; *p* = 0.0356) but was restored in the H_2_-treated group (n = 14, *p* = 0.0035), showing no significant difference from the Sham group (*p* = 0.7980). Freezing behavior was also significantly reduced in the VaD mice (n = 12) compared to the Sham group (n = 13, Tukey’s multiple comparison test; *p* = 0.0255) but it was restored by H_2_ treatment (n = 15, *p* = 0.0036), with no significant difference between the Sham and H_2_-treated VaD mice (*p* = 0.7811). The time spent in the center zone was significantly increased in the VaD mice (n = 12) compared to the Sham group (n = 14, Tukey’s multiple comparison test; *p* = 0.0073), but this increase was reversed by H_2_ treatment compared to the VaD group (n = 17, *p* = 0.0343), with no difference between the Sham and H_2_-treated VaD mice (*p* = 0.7167) (Figure 1F).

In the NORT, mice were allowed to freely explore the objects during the training session. After a rest period, mice were placed back into the chamber containing both a novel object and a familiar object positioned at opposite corners (Figure 1C). This test primarily assessed the mice’s tendency to explore the novel object over the familiar one. In the test results, the distance traveled by the VaD mice (n = 12) was significantly greater than that of the Sham group (n = 13, Tukey’s multiple comparison test; *p* = 0.001) but this was restored in the H_2_-treated VaD mice (n = 12, *p* = 0.0355), with no significant difference between the Sham and H_2_-treated groups (*p* = 0.5254). However, the time spent exploring objects was significantly increased in both VaD (n = 15, Dunn’s multiple comparison test; *p* = 0.0449) and H_2_-treated VaD mice (n = 16, *p* = 0.0005) compared to the Sham mice (n = 12), and no significant difference was detected between the VaD and H_2_-treated VaD mice groups (*p* = 0.5110). Most importantly, the discrimination index, calculated as (time around novel object—time around familiar object)/(time around novel object + time around familiar object), was significantly lower in the VaD group (n = 15) compared to both the Sham (n = 12, Tukey’s multiple comparison test; *p* = 0.0039) and H_2_-treated VaD groups (n = 17, *p* < 0.0001) (Figure 1D). This result indicates that the VaD mice spent a disproportionately longer time exploring the familiar object than the other two groups, suggesting a marked impairment in their memory function. In summary, after BCCAS induced VaD, mice exhibited impaired anxiety and memory function evidenced by altered center zone exploration in the OFT and reduced discrimination ability in the NORT, which were effectively reversed by H_2_ treatment, implying that H_2_ can restore both emotional stability and cognitive performance by mitigating the underlying pathophysiological changes.

### 3.2. H_2_ Attenuates Cellular Pyknosis and Reactive Oxygen Species in the VaD Hippocampus

To assess histological changes following H_2_ treatment in VaD mice, H&E staining was performed (Figure 2A). In the VaD group, pyknotic cells with darker nuclei, indicative of chromatin condensation, were evident, whereas the Sham group’s hippocampal tissue displayed no such features. Notably, this abnormality was reversed in the H_2_-treated group, which exhibited Sham-like, normally condensed chromatin in cells in the hippocampus.

To further evaluate the reactive oxygen species (ROS) scavenging ability of H_2_, (dihydroethidium) DHE staining and its fluorescence microscopy were conducted on both cortical and hippocampal tissues (Figure 2B). Increased DHE accumulation was observed in VaD mice compared to Sham in both the cortex and hippocampus. For quantitative assessment, DHE intensity was analyzed (one-way ANOVA) (Figure 2C). In the cortex, DHE levels were significantly elevated in VaD mice (n = 6) relative to Sham (n = 8, Tukey’s multiple comparison test; *p* = 0.0008) but returned to Sham levels following H_2_ treatment (n = 8, *p* = 0.0001), with no significant difference between the Sham and H_2_-treated groups (*p* = 0.6884). Similarly, in the hippocampus, VaD mice (n = 7) showed significantly higher DHE levels compared to Sham (n = 7, *p* = 0.0403), which were restored by H_2_ treatment (n = 6, *p* = 0.0035), with no significant difference between the Sham and H_2_-treated groups (*p* = 0.4284).

To validate the ROS scavenging ability of H_2_, another assay measuring Malondialdehyde (MDA), which is an organic compound formed by the oxidative degradation of polyunsaturated fatty acids, was conducted using a one-way ANOVA test (Figure 2D). The results showed no significant difference in MDA levels in the cortex among the groups (*p* = 0.0924; n = 6 for all). However, MDA levels were significantly higher in the hippocampus of the VaD group (n = 6) compared to the Sham group (n = 10, Tukey’s multiple comparison test; *p* = 0.0049). This increase was attenuated in the H_2_-treated group (n = 8, *p* = 0.0045) and there was no difference between the Sham and H_2_-treated groups (*p* = 0.9727). Given that the DHE assay measures superoxide generation within cells, while MDA levels reflect cumulative lipid damage over a longer period, these results indicate that the VaD group experienced more substantial oxidative damage in the hippocampus, which was reduced by H_2_ treatment.

### 3.3. H_2_ Enhances Neuronal Survival-Related Protein Expression and Reduces Regulatory T Cell Differentiation

After confirming the neuroprotective role of H_2_ in the hippocampus through cognitive behavioral tests and ROS assays, neuronal survival-related protein markers were assessed in the mouse hippocampus using immunofluorescence microscopy and its quantitative analyses (Kruskal–Wallis test). While BCL2 directly promotes neuronal survival by inhibiting apoptotic pathways and maintaining cellular integrity, IL-4 indirectly supports neuronal survival through its anti-inflammatory and neuroprotective functions that modulate the brain’s immune environment to enhance neuronal health and resilience. The results showed that BCL2 expression in the hippocampus was reduced in the VaD group (n = 5) compared to the Sham group (n = 5; Dunn’s multiple comparison test, *p* = 0.0410) but was restored in the H_2_-treated group (n = 7) to levels comparable to the Sham group (*p* > 0.9999) (Figure 3A,C). In contrast, IL-4 expression in the hippocampus was similar in the Sham (n = 10) and VaD groups (n = 8; Dunn’s multiple comparison test, *p* > 0.9999) (Figure 3B,D). However, the H_2_-treated group (n = 9) exhibited a significantly higher IL-4 expression compared to the Sham group (*p* = 0.0041).

To further assess T cell differentiation in the mouse hippocampus, we quantified populations of CD4^+^ cells, which are crucial for regulating adaptive immune responses; CD4^+^CD25^+^ cells, which represent activated T helper cells; and CD4^+^Foxp3^+^ cells, which indicate regulatory T cells (Tregs) that help suppress excessive or inappropriate immune responses. Flow cytometry analysis was performed for each group and their quantitative measurements were taken (Figure 3E,F). Quantitative analysis of CD4^+^ cells (one-way ANOVA) revealed that their number was significantly decreased in the VaD group (n = 9) compared to the Sham group (n = 8; Dunnett’s multiple comparison test, *p* = 0.0038). However, this decrease was reversed in the H_2_-treated group (*p* = 0.6159). Analysis of CD4^+^CD25^+^ cells (one-way ANOVA) showed that this subset was significantly decreased in the H_2_-treated group (n = 8) compared to the Sham group (n = 9; *p* = 0.0325), whereas no significant difference was observed between the VaD group (n = 8) and Sham group (*p* = 0.7609). Finally, CD4^+^Foxp3^+^ Tregs (one-way ANOVA) were significantly increased in the VaD group (n = 9) compared to the Sham group (n = 9; Dunn’s multiple comparison test, *p* = 0.0208). However, there was no significant difference between the Sham and H_2_-treated groups (n = 8; *p* = 0.0682). In summary, these results suggest that both activated T cells and Tregs are maintained at higher levels in VaD mice and that H_2_ treatment can attenuate this imbalance, helping to restore immunological homeostasis in the brain.

### 3.4. H_2_ Treatment Increases mRNA Expression of Neuronal Regeneration Factors in the Mouse Hippocampus

To investigate the mechanism of H_2_ therapy, mRNA extracted from the hippocampus was analyzed to assess the expression levels of various gene subsets, with target genes selected from well-known markers associated with inflammation and neuro-regeneration. In detail, a one-way ANOVA was performed (post hoc Dunn’s multiple comparison test), focusing on markers related to neurogenesis and synaptic remodeling (Igf2, Gap43), immune regulation (Cd4, Foxp3), synaptic plasticity (*Prkca*, *Prkce*), ROS removal (Gpx5), anti-inflammation (*Il4*, *Il1a*, *Il2ra*), apoptosis (*Bcl2*, *Bad*, *Tgfb1*), and AD-related pathways (App, *Psen1*, *Apoe*) (Figure 4A).

The results demonstrated that neurogenesis and synaptic remodeling were significantly enhanced in the H_2_-treated group, as evidenced by increased mRNA expression of Igf2 (n = 8; *p* = 0.0149) and Gap43 (n = 7; *p* = 0.0400) compared to the Sham group (n = 7 for both analyses). In contrast, no significant differences were observed between the VaD group (n = 8 for Igf2; n = 7 for Gap43) and the Sham group (*p* = 0.2489 for Igf2; *p* = 0.8763 for Gap43).

The mRNA expression of Cd4 and Foxp3 in the H_2_-treated group (n = 6 for both analyses) reflected the same expression pattern observed in the flow cytometry results (Figure 3). No significant differences were detected between the H_2_-treated and Sham groups (n = 7, *p* = 0.4549 for Cd4; n = 6, *p* > 0.9999 for Foxp3). In contrast, the VaD group (n = 8 for Cd4; n = 6 for Foxp3) exhibited a significant decrease in Cd4 expression (*p* = 0.0433) and a significant increase in Foxp3 expression (*p* = 0.0099). For Gpx5, which is responsible for ROS removal, the expression levels in the H_2_-treated group (n = 8) were restored to levels comparable to the Sham group (n = 7; *p* > 0.9999). However, the VaD group (n = 6) exhibited a significant decrease in Gpx5 expression compared to the Sham group (*p* = 0.0395).

The anti-inflammatory factors *Il4* and *Il1a* showed a significant increase in the H_2_-treated group (n = 6 for *Il4;* n = 8 for *Il1a*) compared to the Sham group (n = 7, *p* = 0.0009 for *Il4*; n = 6, *p* = 0.0310 for *Il1a*), whereas no significant changes were observed in the VaD group for both genes. *Il2ra*, another inflammatory factor that expresses CD25 proteins, showed a similar level of expression in the H_2_-treated group (n = 6) as in the Sham group (n = 6; *p* = 0.0570), while it was decreased in the VaD group (n = 6) compared to the Sham group (*p* = 0.0350). Although this result does not align with the findings in Figure 3, it potentially suggests that the total level of CD25 without CD4 co-expression might be maintained at the Sham group level by H_2_ treatment.

The anti-apoptotic *Bcl2* gene was restored in the H_2_-treated group (n = 6) to the level of the Sham group (n = 6; *p* > 0.9999), while it was decreased in the VaD group (n = 6; *p* = 0.0257). The pro-apoptotic *Bad* gene was significantly increased in the VaD group (n = 5) compared to the Sham group (n = 6; *p* = 0.0156) while it was restored in the H_2_-treated group (n = 7; *p* = 0.1352). *Tgfb1*, which promotes neuronal cell survival, was also decreased in the VaD group (n = 7) compared to the Sham group (n = 7; *p* = 0.0070), but it was restored by H_2_ treatment (n = 8; *p* = 0.8386).

App and *Psen1*, which contribute to the regulation of amyloids, were significantly increased in the H_2_-treated group (n = 5 for both analyses) compared to the Sham group (n = 6 for both analyses; *p* = 0.0083 for App; *p* = 0.0168 for *Psen1*), whereas no difference was observed between the Sham and VaD groups (n = 7 for App; n = 6 for *Psen1*; *p* > 0.9999 for both analyses). *Apoe*, which is essential for neuronal repair, was significantly decreased in the VaD group (n = 7) compared to the Sham group (n = 7; *p* = 0.0190) but it was restored in the H_2_-treated group (n = 8; *p* > 0.9999).

A schematic diagram illustrating the intracellular effects of H_2_ based on the results of flow cytometry and mRNA expression analysis is presented (Figure 4B). The illustration depicts differentiated Treg cells and neuronal cells in the VaD brain based on the findings of this study combined with existing knowledge. In an ROS-excessive environment, GPX5 is activated to mitigate oxidative stress, BAD is activated to promote apoptosis, and cell survival-related BCL2 is attenuated through upstream PKC activation. PKCε is also thought to be activated, while IL-4, GAP43, and APOE are attenuated under such conditions.

H_2_ is hypothesized to inhibit ROS activity, thereby promoting the expression of GPX5, BCL2, GAP43, PSEN1, and APP in neuronal cells, contributing to neuroprotection. At the same time, H_2_ inhibits the expression of BAD, PKCα, and PKCε, attenuating cell death. H_2_ also enhances IGF2 expression, which supports synaptic plasticity and neurogenesis. Furthermore, H_2_ upregulates cytokines such as TGFβ1, IL-4, and IL-1a, promoting immune regulation and anti-inflammatory responses. Additionally, H_2_ increases APOE expression, critical for amyloid beta regulation. In T cells, H_2_ is proposed to inhibit differentiation into Treg cells while restoring CD4^+^ cell levels to normal, potentially due to immune suppression and the inhibition of T cell differentiation.

## 4. Discussion

The present study demonstrates that H_2_ treatment effectively restores cognitive and emotional functions in a VaD mouse model induced by BCCAS. Various methods exist for creating VaD models, each with unique strengths and limitations. For instance, the multiple infarct and thromboembolism model, involving the injection of micro-emboli into the internal carotid artery, is considered clinically relevant but fails to sustain long-term deficits, making it unsuitable for this study, which aimed to evaluate treatment responses over an extended period [27].

Another model, unilateral CCA occlusion, results in only mild cerebral blood flow reduction, producing weak symptoms and limited pathology, and was therefore deemed inappropriate for assessing the efficacy of our treatment [28,29]. In contrast, the BCCAS model predominantly induces white matter lesions in the corpus callosum and enhances the inflammatory response [30]. Although this model can sometimes lead to only mild cerebral blood flow reduction, we addressed this limitation by using a smaller diameter coil (0.16 mm) instead of the standard 0.18 mm coil. This modification enabled the establishment of a VaD model that exhibited measurable anxiety-impaired behavior and memory deficits, as demonstrated through cognitive behavioral tests (Figure 1F). These findings parallel clinical observations in human VaD patients, who experience a gradual reduction in anxiety during the advanced stages of the disease and exhibit significant dementia-related impairments in learning and memory following a stroke [31,32].

Our current model also represents an improvement over our previous approach. Previously, we induced ischemia–reperfusion injury by temporarily clamping both CCAs and then releasing them after a set duration [22]. While this method generated degenerative changes in the brains of mice, it produced a singular, transient injury event. In the current study, we established permanent stenosis of the CCAs, thus creating a sustained hypoperfusion environment. This model more closely mimics the chronic hypoperfusion observed in patients with atherosclerotic internal carotid artery stenosis. Although the outcomes appear to be like those of the previous model, the underlying principle differs significantly. Here, we demonstrated the effectiveness of H_2_ in delaying cognitive decline in a persistent, rather than transient, environment.

To assess the effect of H_2_ treatment, we compared mice that drank H_2_-rich water (HRW) with those that did not. Compared to the VaD group, the H_2_-treated mice displayed anxiety and memory performances similar to those of the Sham group. This clinical improvement aligns with previous studies, suggesting that H_2_ provides therapeutic benefits in VaD models [22]. Results from the NORT revealed that VaD mice exhibited increased movement distance, potentially reflecting characteristic symptoms associated with dementia such as irritability [33]. In contrast, H_2_-treated mice demonstrated improved exploratory behavior and a stronger preference for the novel object, indicating a notable enhancement in learning and memory abilities due to H_2_ treatment. Interestingly, this behavioral pattern differed from that of the Sham group, suggesting unique cognitive recovery dynamics in the H_2_-treated mice (Figure 1D).

Histological and biochemical analyses further revealed that H_2_ reduces oxidative stress in the VaD brain. H&E staining showed reversal of pyknotic changes in hippocampal neurons, while DHE staining confirmed reduced ROS levels in both the cortex and hippocampus of H_2_-treated mice (Figure 2). Additionally, MDA assays demonstrated attenuated lipid peroxidation in the H_2_-treated group. At the molecular level, H_2_ treatment restored Gpx5 mRNA expression, a critical ROS scavenger, to near-Sham levels, thereby augmenting antioxidant defenses and minimizing oxidative stress-induced neuronal damage (Figure 4A). These results are in line with previous work showing that H_2_ can serve as a therapeutic agent by reducing ROS [20,21,22,34].

H_2_ treatment also influenced apoptotic and survival signaling pathways (Figure 4A). In VaD mice, the pro-apoptotic gene *Bad* was upregulated and the anti-apoptotic gene *Bcl2* was downregulated, indicating heightened neuronal vulnerability. H_2_ treatment restored *Bcl2* and suppressed *Bad*, shifting the balance toward cell survival [35]. *Tgfb1*, a cytokine that promotes neuronal survival, was also normalized by H_2_ treatment, suggesting improved neuronal integrity and survival through the precise regulation of microglial activation [36]. Additionally, H_2_ enhanced the expression of neurogenesis and synaptic remodeling markers such as Igf2 and Gap43, suggesting improved synaptic plasticity and regeneration [37]. It also upregulated App and *Psen1*, associated with amyloid regulation, and restored *Apoe* expression, a key factor in neuronal repair [38,39,40,41,42]. Altogether, these changes suggest that H_2_ supports structural and functional recovery and may impact pathways relevant to Alzheimer’s disease.

Immunologically, H_2_ promoted an anti-inflammatory environment. Anti-inflammatory cytokines *IL-4* and *IL-1a* were significantly elevated and *Il2ra* (encoding CD25) expression was restored, indicating an improved immunological balance. Flow cytometry revealed that H_2_ normalized CD4^+^ T cell populations and reduced the proportion of Tregs (CD4^+^Foxp3^+^) elevated in VaD mice. Thus, H_2_ modulates immune responses, maintaining homeostasis and limiting excessive inflammation. The reduction in activated T cells is indicative of a non-toxic brain environment marked by cellular equilibrium [43]. Accordingly, the flow cytometry results likely represent a low inflammatory state in the brain, eliminating the need for toxic mechanisms to remove cells (Figure 3E,F).

A schematic diagram (Figure 4B) illustrates the hypothesized mechanisms of H_2_ action in VaD cases. Under ROS-excessive conditions, GPX5 scavenges ROS, and apoptotic markers like BAD and BCL2 mediate neuronal fate. PKCα and PKCε regulate these apoptotic processes, while IL-4, GAP43, and APOE expression is diminished under oxidative stress. H_2_ treatment is thought to counter these processes, restoring GPX5, BCL2, GAP43, PSEN1, and APP, while inhibiting BAD, PKCα, and PKCε. H_2_ also potentially promotes IGF2 for neurogenesis and synaptic plasticity and elevates anti-inflammatory cytokines such as TGFβ1, IL-4, and IL-1a. Together, these adjustments improve cognitive and emotional functions by restoring both neuronal and immunological homeostasis.

Nonetheless, our approach presents several limitations. The administration of H_2_ poses significant challenges. Although intraperitoneal injection allows for precise dosage control, it can induce stress in animals, potentially confounding behavioral outcomes. While H_2_ gas inhalation serves as an alternative, its technical complexity led us to opt for oral administration via HRW. However, despite consistently observing a significant reduction in hippocampal ROS levels, the standard error of the mean in the DHE assays indicates variability among individual mice, likely attributable to irregular oral intake (Figure 2C,D). While this method simplifies treatment delivery, it complicates the accurate quantification of individual intake. Future studies could benefit from indirect analytical techniques, such as measuring MDA concentrations in blood, to assess H_2_ consumption per animal and determine its influence on treatment efficacy.

Furthermore, our VaD model exhibits certain limitations. Although we have advanced the design to overcome previous challenges in mouse model development, this model results in high inter-animal variance due to inherent vascular conditions, rendering some animals unsuitable for use as VaD models and introducing outliers. Additionally, the coil insertion procedure is technically demanding, requiring highly skilled personnel to perform the surgery effectively. Moreover, the gold micro-coil material is not fully biodegradable, raising concerns about potential long-term effects on the mice’s physiology while the coil remains in their bodies. Further considerations of the methodology for establishing the model are required for future advanced studies.

Additionally, as MDA and DHE levels measured in the cortex vary depending on the assay method used, careful selection of ROS measurement techniques is crucial to ensure accurate and reliable results. To address this issue, we plan to implement multiple ROS measurement methods to achieve more accurate and reliable quantification of ROS levels in animals.

In conclusion, this study presents a refined VaD mouse model that more closely mimics chronic hypoperfusion and demonstrates the protective role of H_2_ treatment in cognitive and emotional functions. H_2_ attenuates oxidative stress, reduces apoptosis, enhances neurogenesis and synaptic plasticity, and maintains immunological balance. Although further research is needed to optimize H_2_ dosing and elucidate its precise molecular mechanisms, our findings suggest that H_2_ may serve as a promising therapeutic approach for VaD.

## 5. Conclusions

This study demonstrates the therapeutic efficacy of molecular hydrogen (H_2_) in improving cognitive and emotional deficits in a refined vascular dementia (VaD) mouse model induced by bilateral common carotid artery stenosis (BCCAS). H_2_ treatment effectively restored anxiety and memory performances, reversed oxidative stress, and mitigated neuronal damage by enhancing antioxidant defenses, reducing apoptosis, and promoting synaptic plasticity and neurogenesis. Furthermore, H_2_ modulated immune responses, restoring immunological balance and reducing excessive inflammation. These findings underscore the potential of H_2_ in addressing key pathophysiological processes in VaD cases, including oxidative damage, neuronal vulnerability, and immune dysregulation.

While this study highlights the promise of H_2_ therapy, limitations such as variability in oral intake and challenges in precise dosage delivery require further refinement. Future research should focus on optimizing administration methods, implementing robust reactive oxygen species (ROS) measurement techniques, and exploring the molecular pathways underlying H_2_’s protective effects. Overall, our findings suggest that H_2_ is a multifaceted and promising therapeutic strategy for VaD, offering potential applications in broader neurodegenerative and cognitive disorders.

## Figures and Tables

**Figure 1 antioxidants-14-00111-f001:**
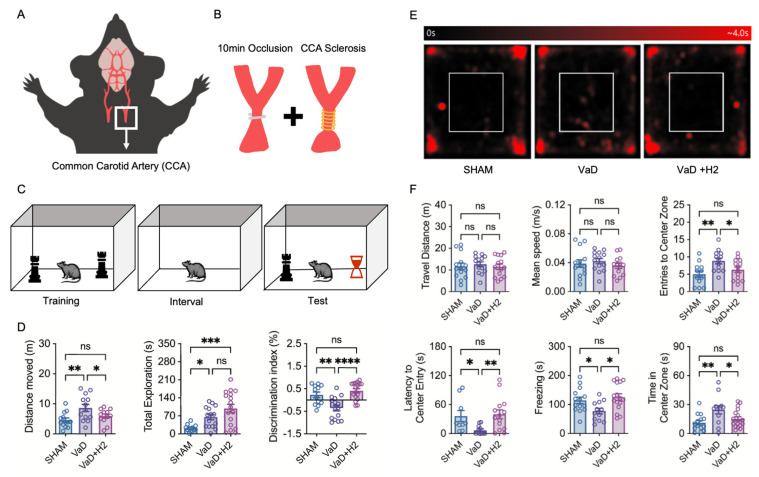
A mouse model of vascular dementia (VaD) and its cognitive and behavioral assessments using novel object recognition (NOR) and open field (OF) tests. (**A**,**B**) Surgical procedures for establishing the VaD model, including bilateral common carotid artery (BCCA) occlusion (10 min) and sclerosis, are illustrated. (**C**) In the NORT procedure, following a training session with identical objects, the mouse is placed into an empty chamber during the interval and then tested with both a familiar and a novel object to assess recognition memory. (**D**) In the NORT, VaD mice traveled greater distances within the chamber, which was restored by H_2_ treatment. On the other hand, object exploration was greatest in the H_2_-treated group. Object discrimination, measured by the discrimination index, was impaired in VaD mice and restored to near Sham levels by H_2_ treatment. ns = not significant; * *p* < 0.05; ** *p* < 0.01; *** *p* < 0.001; **** *p* < 0.0001. (**E**) A heat map of OFT results shows the mice’s movement, with red indicating prolonged time spent in a single location. (**F**) Although no differences in travel distance or mean speed were observed between groups, center zone entries were significantly higher in VaD mice. Additionally, VaD mice exhibited reduced latency to center entry and decreased freezing behavior, alongside increased time spent in the center zone. ns = not significant; * *p* < 0.05; ** *p* < 0.01.

**Figure 2 antioxidants-14-00111-f002:**
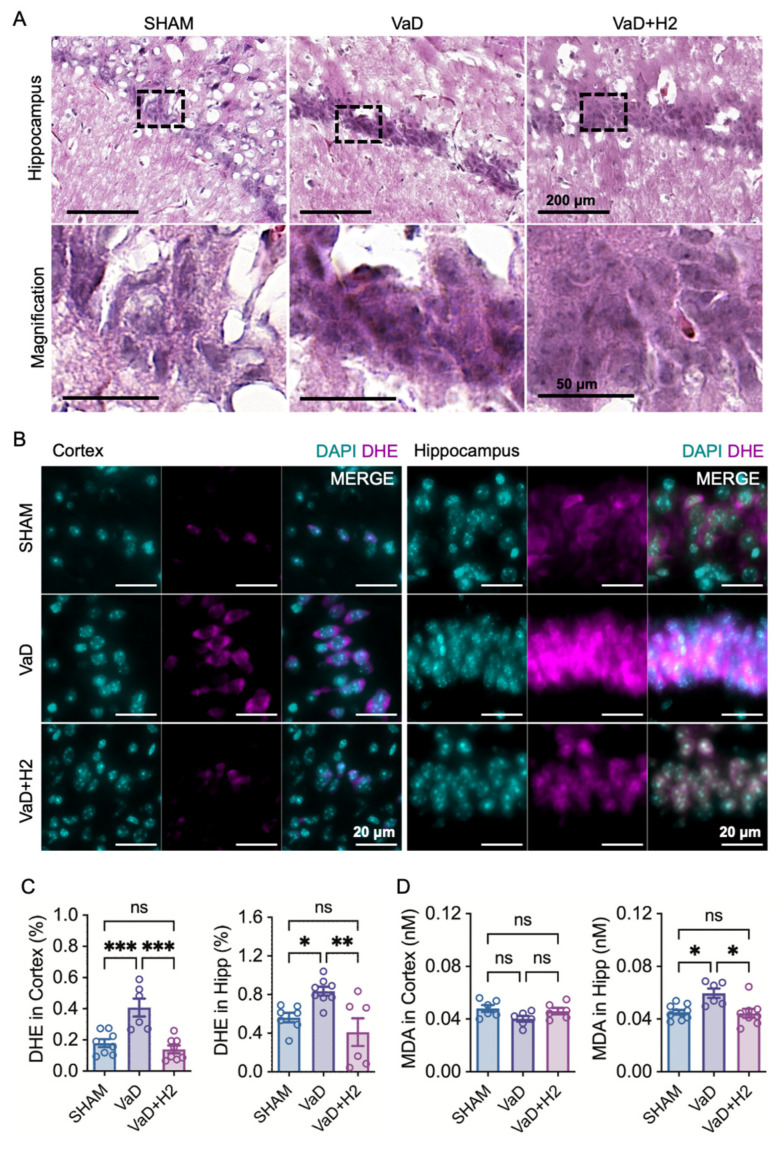
Histological configuration of the VaD mouse model and evaluation of reactive oxygen species (ROS) using DHE fluorescence and Malondialdehyde (MDA) concentration analyses. (**A**) H&E-stained images show condensed chromatin with pyknotic features in the VaD group. (**B**) Representative fluorescence microscopy images of DHE staining are presented and (**C**) quantification of DHE fluorescence intensity indicates significantly elevated ROS levels in both the cortex and hippocampus of VaD mice, which are attenuated by H_2_ treatment. ns = not significant; * *p* < 0.05; ** *p* < 0.01; *** *p* < 0.001. (**D**) MDA concentration measurements reveal a marked increase in hippocampal MDA levels but no difference in the cortex of VaD mice. ns = not significant; * *p* < 0.05.

**Figure 3 antioxidants-14-00111-f003:**
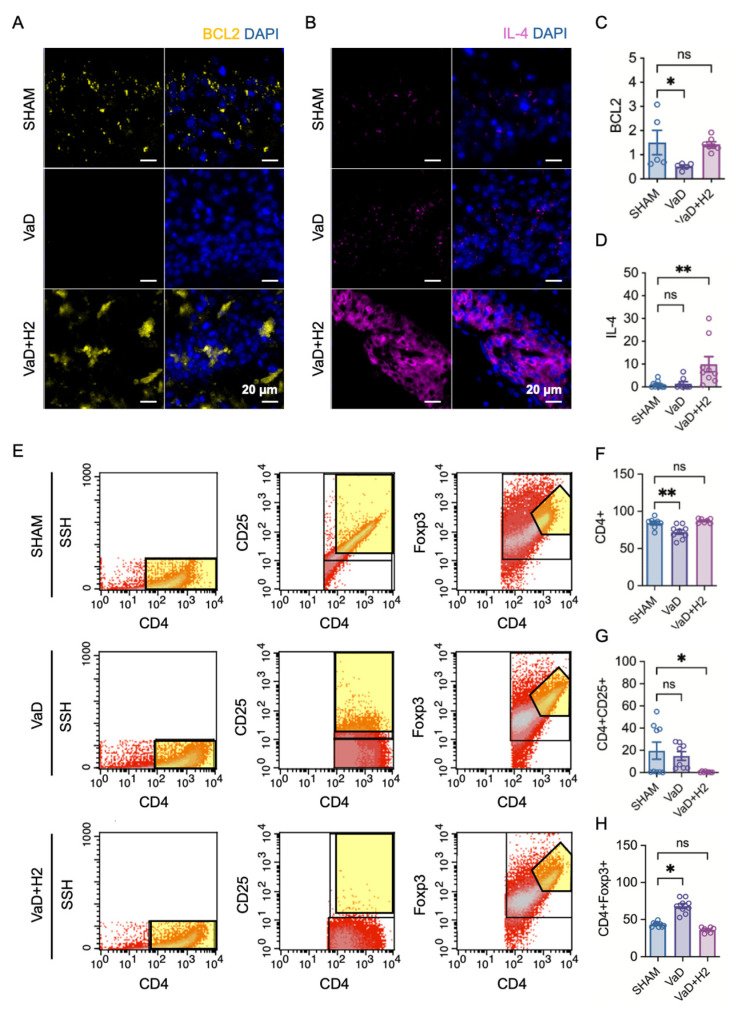
Immunofluorescence images of Bcl-2 and IL-4 expression in the hippocampus and flow cytometry analyses of Treg pathway activation. (**A**,**B**) Representative immunofluorescence images show Bcl-2 (yellow) and IL-4 (magenta) expression, with nuclei stained by DAPI (blue). (**C**,**D**) Quantitative analyses of Bcl-2 and IL-4 expression levels are presented as bar graphs. ns = not significant; * *p* < 0.05; ** *p* < 0.01. (**E**) Flow cytometry gating strategy for identifying CD4^+^, CD4^+^CD25^+^, and CD4^+^Foxp3^+^ T cells (indicated by yellow regions). (**F**–**H**) Flow cytometry results demonstrate that CD4^+^ cell counts were significantly decreased in the VaD group but restored by H_2_ treatment. In contrast, CD4^+^CD25^+^ cells were significantly lower in the H_2_-treated group, while CD4^+^Foxp3^+^ cells were significantly higher in the VaD group than in the other groups. ns = not significant; * *p* < 0.05; ** *p* < 0.01.

**Figure 4 antioxidants-14-00111-f004:**
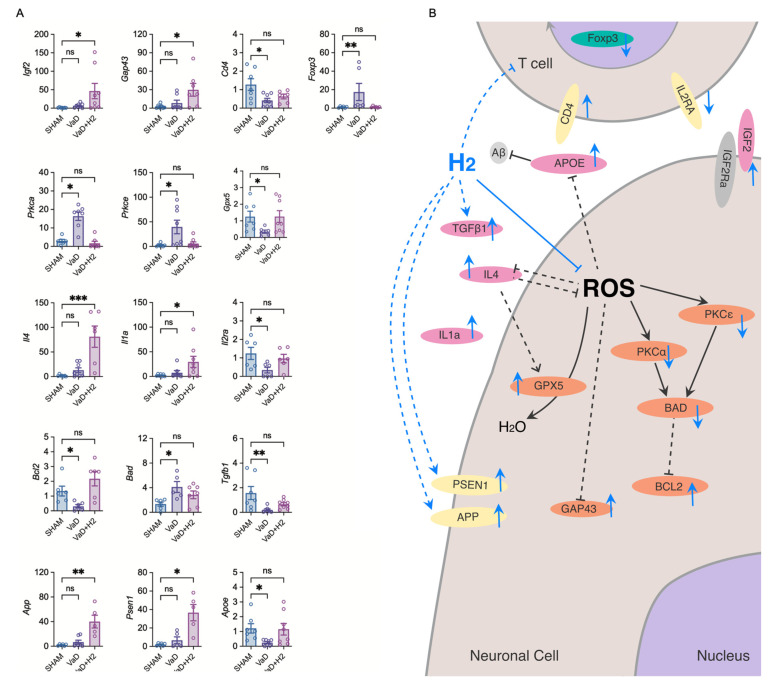
Analysis of mRNA expression in the VaD group related to inflammation, regulatory T cells (Tregs), apoptosis, neuronal regeneration, and Alzheimer’s disease. (**A**) The fold changes in mRNA levels of the selected target genes. ns = not significant; * *p* < 0.05; ** *p* < 0.01; *** *p* < 0.001. (**B**) A signaling pathway diagram illustrating the effects of molecular hydrogen. Black arrows represent established mechanisms of action, blue arrows highlight findings from this study, and dashed lines denote indirect effects on the targets. Gray: known mechanism; orange: cytoplasmic proteins; yellow: transmembrane proteins; pink: cytokines; green: nuclear proteins.

## Data Availability

The data that support the findings of this study are available from the corresponding author upon request.

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
