# Peer review of "Molecular Hydrogen Modulates T Cell Differentiation and Enhances Neuro-Regeneration in a Vascular Dementia Mouse Model"

_antioxidants, 2025, doi:10.3390/antiox14010111_

Round 1
Reviewer 1 Report
Recommendations:
1) Revise the Introduction: Increase emphasis on the pathophysiology of VaD, underlining the differences with other types of dementias. State in a clear manner why hydrogen rich water treatment can be considered as a possible treatment for VaD.
2) Enhance the Methodology Section: Elaborate on the BCCAS model, the technical aspect of it. The authors should also explain in more detail how the model effectively depicts the major characteristics of VaD in human beings.
Mechanisms of Action of Hâ‚‚:
- Although the mechanisms through which molecular hydrogen (Hâ‚‚) may be effective in the treatment of VaD are discussed, the specific details of how Hâ‚‚ works at a molecular level to produce these benefits are not elaborated on in sufficient detail. This is where more details about how Hâ‚‚ interacts with the cells would be useful.
Author Response
Comment 1: Revise the Introduction: Increase emphasis on the pathophysiology of VaD, underlining the differences with other types of dementias. State in a clear manner why hydrogen rich water treatment can be considered as a possible treatment for VaD.
Response 1: We expanded the discussion on the role of molecular hydrogen in the treatment of vascular dementia (VaD) in the introduction. Specifically, we contrasted VaD with Alzheimer's disease to highlight the critical importance of mitigating reactive oxygen species (ROS) in the VaD-affected brain through the application of molecular hydrogen (Line 72-79).
Recently, molecular hydrogen (Hâ‚‚) has emerged as a promising therapeutic agent [19,20]. Studies have shown that Hâ‚‚ has antioxidative, antiapoptotic, and an-ti-inflammatory effects in various conditions, including cancer, lung damage, and neu-rodegenerative diseases, by neutralizing ROS, which cause damage to DNA, proteins, and cells [21–23]. Given the significant role of reactive oxygen species in exacerbating neurological damage, Hâ‚‚ presents a potential strategy for mitigating oxidative stress in dementia-related conditions. VaD differs in its etiology from AD, as it occurs due to neurovascular dysfunction that induces ischemia or reperfusion, which can greatly in-crease the reactive oxygen species (ROS) that harm tissue. In contrast, AD primarily occurs through the accumulation of amyloid beta plaques. Therefore, managing the excessive accumulation of ROS in the brains affected by VaD constitutes a more critical challenge compared to AD.
Comment 2: Enhance the Methodology Section: Elaborate on the BCCAS model, the technical aspect of it. The authors should also explain in more detail how the model effectively depicts the major characteristics of VaD in human beings.
Response 2: We enhanced the description of the model by adding additional sentences for greater clarity. Furthermore, we incorporated design features that mimic the characteristics of vascular dementia (VaD) in humans, as detailed below (Line 133-147).
Anesthesia was induced in mice through inhalation of 5% isoflurane in 1 L of oxygen using an animal anesthesia machine (L-PAS-01D, LMS Korea, Pyeongtaek, Korea). Full anesthesia was achieved within 3 min. During the surgical procedure, anesthesia was maintained with 3% isoflurane in 600mL of oxygen. A vertical midline incision was made along the ventral neck of each mouse, and the soft tissues and muscles were carefully dissected to sequentially isolate both common carotid arteries (CCAs). Both CCAs were occluded for 10 min using bulldog clips. To mitigate the risk of mortality due to reper-fusion injury, the clips were removed at 3-min intervals. Subsequently, a 0.16 mm coil (SWPA 0.16, Sawane SPRING CO., LTD., Hamamatsu, Japan) was wrapped around each common carotid artery to induce stenosis. After suturing the incision, the mice were housed individually until consciousness was restored, after which they were returned to their original cages. All procedures were completed within 30 min. This model was de-signed to mimic the conditions of VaD, characterized by reperfusion-induced damage and continuous cerebral hypoperfusion, by employing an occlusion method aligned with BCCAS operational protocols.
Commend 3: Mechanisms of Action of Hâ‚‚: Although the mechanisms through which molecular hydrogen (Hâ‚‚) may be effective in the treatment of VaD are discussed, the specific details of how Hâ‚‚ works at a molecular level to produce these benefits are not elaborated on in sufficient detail. This is where more details about how Hâ‚‚ interacts with the cells would be useful.
Response 3:
We added sentences for the detailed elaboration of the effect of molecular hydrogen (Line 80-87).
In detail, Hâ‚‚ exerts its therapeutic effects on vascular dementia (VaD) through mul-tiple cellular interactions at the molecular level. Primarily, Hâ‚‚ acts as a selective anti-oxidant, neutralizing harmful reactive oxygen species such as hydrogen peroxide (H2O2) and peroxynitrite (ONOO−), thereby reducing oxidative stress and preventing neuronal damage. It also modulates inflammatory responses by suppressing pro-inflammatory cytokines and inhibiting transcription factors like NF-κB, which helps mitigate neural inflammation associated with VaD. Hâ‚‚ further enhances the cell’s antioxidant de-fenses by activating the Nrf2 pathway, leading to increased expression of antioxidant enzymes.
Reviewer 2 Report
The study focuses on the potential therapeutic effect of H2. Using a vascular dementia mouse model the authors demonstrated that molecular hydrogen could mitigate cognitive and immunological disturbances triggered by bilateral common carotid artery stenosis. In general, the study design is logical and presented data are comprehensive and of satisfactory quality. Various biomarkers related to inflammation and immune regulation (e.g., IL4), oxidative stress (e.g., DHE), apoptosis and cell survival (e.g., BCLS) and Alzheimer’s Disease-related pathways were monitored between the three mouth groups, SHAM, VaD and VaD+H2. The comparison clearly highlights the potential therapeutic benefit of H2. Below please find a few minor comments for consideration.
1. Section 2. Provide approved animal study protocol.
2. Hâ‚‚ is believed to selectively react with certain reactive oxygen species (ROS), such as hydroxyl radicals. Is the oxidative stress observed in the VaD group primarily due to hydroxyl radicals, or are other ROS, which might not interact with Hâ‚‚, also significant contributors?
3. Line 228, the authors mention a 12-week treatment duration. Is there a specific rationale for this choice? Were shorter or longer treatment durations tested, and if so, how did the results compare?
4. The authors investigated several pathways modulated by H2 but it remains unclear how H2 directly influenced Foxp3 expression in Tregs. Did the authors observe any off-target effect of H2?
5. The authors carried out behavioral tests (e.g., OFT, NORT) to assess cognitive and emotional recovery in mice. Are the observations consistent with human VaD patients?
The study focuses on the potential therapeutic effect of H2. Using a vascular dementia mouse model the authors demonstrated that molecular hydrogen could mitigate cognitive and immunological disturbances triggered by bilateral common carotid artery stenosis. In general, the study design is logical and presented data are comprehensive and of satisfactory quality. Various biomarkers related to inflammation and immune regulation (e.g., IL4), oxidative stress (e.g., DHE), apoptosis and cell survival (e.g., BCLS) and Alzheimer’s Disease-related pathways were monitored between the three mouth groups, SHAM, VaD and VaD+H2. The comparison clearly highlights the potential therapeutic benefit of H2. Below please find a few minor comments for consideration.
1. Section 2. Provide approved animal study protocol.
2. Hâ‚‚ is believed to selectively react with certain reactive oxygen species (ROS), such as hydroxyl radicals. Is the oxidative stress observed in the VaD group primarily due to hydroxyl radicals, or are other ROS, which might not interact with Hâ‚‚, also significant contributors?
3. Line 228, the authors mention a 12-week treatment duration. Is there a specific rationale for this choice? Were shorter or longer treatment durations tested, and if so, how did the results compare?
4. The authors investigated several pathways modulated by H2 but it remains unclear how H2 directly influenced Foxp3 expression in Tregs. Did the authors observe any off-target effect of H2?
5. The authors carried out behavioral tests (e.g., OFT, NORT) to assess cognitive and emotional recovery in mice. Are the observations consistent with human VaD patients?
Author Response
Comment 1: Section 2. Provide approved animal study protocol.
Response 1: We included the statement, “This study was approved by the Institutional Animal Care and Use Committee (IACUC) at Korea University Medical Center (Approval No. KOREA-2021-0176),” in Section 2.1 Animals, lines 114-115.
Comment 2: Hâ‚‚ is believed to selectively react with certain reactive oxygen species (ROS), such as hydroxyl radicals. Is the oxidative stress observed in the VaD group primarily due to hydroxyl radicals, or are other ROS, which might not interact with Hâ‚‚, also significant contributors?
Response 2: We sincerely appreciate your insightful question and the comprehensive context it provides. Vascular dementia (VaD) is widely recognized to involve significantly elevated levels of reactive oxygen species (ROS), including superoxide anions, hydrogen peroxide, hydroxyl radicals, peroxynitrite, and singlet oxygen. In our study, the DHE assay results specifically indicate an increase in superoxide anions, while the MDA assay may reflect the presence of one or more of these ROS capable of inducing lipid peroxidation.
Molecular hydrogen (Hâ‚‚) is known to selectively interact with hydroxyl radicals and peroxynitrite, the primary ROS implicated in oxidative damage within VaD. Although the precise ROS targeted by hydrogen-rich water (HRW) in our study have yet to be fully elucidated, the observed reductions in both DHE and MDA assay results, along with increased GPX5 activity, support the hypothesis that Hâ‚‚ treatment effectively ameliorates the ROS-related environment in VaD mice under disease conditions. Specifically, the decrease in DHE fluorescence intensity suggests that Hâ‚‚ may also exert secondary effects that reduce superoxide anion levels, potentially through indirect mechanisms such as enhancing endogenous antioxidant defenses.
While these interpretations are currently based on our hypotheses and the existing data, we highly value your question and recognize the necessity for further investigation. In future studies, we aim to precisely delineate the specific ROS subtypes affected by Hâ‚‚ application. This will involve employing additional targeted assays and mechanistic studies to deepen our understanding of how Hâ‚‚ exerts its therapeutic effects in VaD, thereby reinforcing the evidence for its potential use as an antioxidant therapy in this context.
Comment 3: Line 228, the authors mention a 12-week treatment duration. Is there a specific rationale for this choice? Were shorter or longer treatment durations tested, and if so, how did the results compare?
Response 3: We also tested an eight-month vascular dementia (VaD) model based on the premise that VaD develops progressively over an extended period. However, this model presented significant challenges, notably a high mortality rate throughout the eight-month duration. In contrast, many animal models used in VaD research adopt a 12-week induction period. We posit that a 12-week timeframe provides an optimal balance, allowing researchers to observe distinct neuropathological changes characteristic of VaD while maintaining relatively low mortality rates.
Consequently, we selected the 12-week VaD model for our studies, ensuring that the hydrogen treatment protocol was consistent with the setup applied to the VaD induction. Although some studies utilize shorter model durations, our objective was to investigate the long-term effects of hydrogen (Hâ‚‚) treatment. Recognizing the validity of alternative approaches, we plan to incorporate varied treatment time points in our upcoming research. Specifically, we will implement hydrogen-rich water (HRW) interventions at intervals of one week, three months, and eight months to comprehensively assess the therapeutic potential of Hâ‚‚ across different stages of VaD progression.
Comment 4: The authors investigated several pathways modulated by H2 but it remains unclear how H2 directly influenced Foxp3 expression in Tregs. Did the authors observe any off-target effect of H2?
Response 4: We sincerely appreciate your insightful question. In this study, we established a chronic vascular dementia (VaD) model that retains not only neurovascular damage but also broader dysfunctions, including immune response, cellular control, apoptosis, and neuroprotection. By administering molecular hydrogen (Hâ‚‚) over an extended period, we anticipated that the results would reflect not only the direct effects of Hâ‚‚ but also overall improvements in the brain environment.
Our long-term Hâ‚‚ treatment was expected to enhance various aspects of brain function, potentially influencing the regulatory T cells (Tregs) pathway. Specifically, we aimed to assess how the improved brain environment might affect Tregs and to identify which aspects of brain function are modulated by Hâ‚‚ regulation. Consequently, we observed a recovery of Foxp3 expression to levels comparable to the sham group. However, further studies are necessary to confirm the effects of hydrogen-rich water (HRW) on Tregs. These investigations should be conducted at the in vitro level to specifically elucidate the mechanisms involved.
Currently, we are unable to perform cell culture experiments, which limits our ability to thoroughly examine the off-target effects of Hâ‚‚ and to understand the precise mechanisms by which Hâ‚‚ regulates the Tregs pathway in the VaD model. Moving forward, we plan to enhance our study by incorporating in vitro data in future study. This will allow us to confirm the off-target effects of Hâ‚‚ and to clarify how Hâ‚‚-mediated regulation of the Tregs pathway contributes to the therapeutic effects observed in the Hâ‚‚-treated VaD model.
By addressing these aspects in future research, we aim to provide a more comprehensive understanding of the multifaceted benefits of Hâ‚‚ treatment in VaD, thereby reinforcing its potential as a viable antioxidant therapy for improving brain health in this context.
Comment 5: The authors carried out behavioral tests (e.g., OFT, NORT) to assess cognitive and emotional recovery in mice. Are the observations consistent with human VaD patients?
Response 5: Yes, to improve clarity, we have added a thorough supporting discussion with relevant citations in the discussion section (lines 481-484).
Another model, unilateral common carotid artery (CCA) occlusion, results in only mild cerebral blood flow reduction, producing weak symptoms and limited pathology, and was therefore deemed inappropriate for assessing the efficacy of our treatment [27,28]. In contrast, the BCCAS model predominantly induces white matter lesions in the corpus callosum and enhances the inflammatory response [29]. Although this model can sometimes lead to only mild cerebral blood flow reduction, we addressed this limitation by using a smaller diameter coil (0.16 mm) instead of the standard 0.18 mm coil. This modification enabled the establishment of a VaD model that exhibited measurable anxiety-impaired behavior and memory deficits, as demonstrated through cognitive behavioral tests (Figure 1F). These findings parallel clinical observations in human VaD patients, who experience a gradual reduction in anxiety during the advanced stages of the disease and exhibit significant dementia-related impairments in learning and memory following a stroke.
Round 2
Reviewer 1 Report
This research presents significant preliminary results regarding the therapeutic potential of hydrogen-rich water treatment in a mouse model of vascular dementia. The study regarding the treatment's influence on T cell differentiation and subsequent neuroregeneration marks a notable progress in our understanding of molecular hydrogen's contribution to reducing cognitive decline and improving the emotional health linked with this important issue. The authors have updated the manuscript, responding to all the issues noted in the review. I believe the manuscript is suitable for publication in the present form.
I have no further comments on the manuscript.